# Analyzing Consumer Perception on Quality and Safety of Frozen Foods in Emerging Economies: Evidence from Albania and Kosovo

**DOI:** 10.3390/foods11091247

**Published:** 2022-04-26

**Authors:** Arbenita Hasani, Elena Kokthi, Oltjana Zoto, Kaltrina Berisha, Iliriana Miftari

**Affiliations:** 1Department of Food Technology with Biotechnology, Faculty of Agriculture and Veterinary, University of Prishtina “Hasan Prishtina”, 10000 Prishtina, Kosovo; arbenita.hasani@uni-pr.edu (A.H.); kaltrina.berisha@uni-pr.edu (K.B.); 2Department of Agriculture Economics, Faculty of Agriculture and Veterinary, University of Prishtina “Hasan Prishtina”, 10000 Prishtina, Kosovo; 3Food Science Department, College of Agriculture and Life Sciences, Cornell University, Ithaca, NY 14850, USA; 4Department of Food and Biotechnology, Faculty of Biotechnology and Food, Agriculture University of Tirana, 9302 Tirana, Albania; ekokthi@ubt.edu.al; 5Department of Economy and Rural Development Policy, Faculty of Economy and Agribusiness, Agriculture University of Tirana, 9302 Tirana, Albania; ozoto@ubt.edu.al; 6Department of Nutrition, Doctoral School of Food Science, Hungarian University of Agriculture and Life Science, 2100 Gödöllő, Hungary

**Keywords:** frozen food, quality, safety, consumer perception, Balkans

## Abstract

Freezing technology is one of the most well established long-term preservation techniques for producing high-quality, nutritious foods with prolonged shelf-life. Frozen foods (FFs) are a significant section of the global food market experiencing rapid growth. It also represents an alternative to small producers in developing countries to add value to their products in a competitive market. However, unfairly, FFs are often perceived as less qualitative than fresh produce, although studies have shown that some FFs have higher nutritional values than fresh products. This study’s aim is to analyze consumers’ perceptions in the two Balkan countries towards FFs. A total of 380 questionnaires were completed in both countries (182 in Kosovo and 198 in Albania). Consumers’ perceptions towards FFs were measured through eleven items using a five-point Likert scale. The items addressed issues related to the quality and safety of FFs, information on FFs, and the impact of origin on the perception of FFs. The differences between populations were tested with the t-test and correlation analysis with the bootstrapping method for sociodemographic factors. The results show that Kosovo consumers generally show a higher positive attitude toward FFs than Albanian consumers. Albanian consumers prefer fresh foods over FFs. The lack of trust in food safety institutions was expressed with concern for the conditions of the frozen chain applied both on the imported and domestic frozen products. Similarly, the findings show that Albanian consumers are willing to pay more than the baseline price to obtain fresh products instead of frozen compared with Kosovo consumers. Additional studies are needed to explore whether the lack of trust in food safety institutions inhibits the successful development of FFs in Albania and Kosovo. In both countries, responsible authorities should help consumers to have a more profound knowledge of the quality of FFs and boost these activities to increase farmers’ incomes and play an active role in reducing food loss and waste.

## 1. Introduction

Knowledge of consumer behavior towards a category of products is necessary for research and development during the development of new products generally, and food products in particular, in all phases of a product cycle [1].

FFs comprise an important section of the global food market, experiencing rapid growth from year to year. A growth rate of 6.5% is estimated, and the market share will be USD 224.2 billion by 2025 [2]. Technical and innovation considerations will significantly impact the future of FFs. Population increase, personal wealth, the relative cost of other types of food, changes in tastes and preferences, and technological breakthroughs in freezing technologies are all elements that will influence the future of freezing technology. Indeed, the demand for ready-to-eat products, semi-ready-to-eat products, and FF products has increased, as people have less free time to cook, especially with the increasing number of employed women [1]. Developing countries follow the same trend. The proportion of fresh food preserved by freezing is highly related to economic development in a society. As countries become wealthier, their demand for high-valued commodities increases, primarily due to income, with increases in the consumption of high-value commodities in developing countries [3].

Today, freezing is the only large-scale process that connects the seasons and variations in the availability and demand of raw materials, such as meat, fish, butter, fruits, and vegetables. Furthermore, it enables the transport of enormous quantities of food across long distances [4]. However, frozen products are often seen as second-hand and lower quality than fresh produce [2,3]. Despite this perception, studies have shown that several frozen products have more nutritional value than any other form of preservation and, in some cases, even higher nutritional values than fresh products [3,4,5,6,7]. The freezing method maintains the quality of agricultural goods for extended periods [8]. In terms of sensory and nutritional characteristics, freezing is usually preferable to canning and dehydration as a long-term preservation option for fruits and vegetables [1]. Additionally, freezing is considered one of the best methods for long-term food preservation, since chemical reactions and microbial growth reduce at low temperatures [8]. The freezing process combines the advantages of low temperatures, where microbes cannot grow, chemical reactions are slowed, and cellular metabolic responses are delayed [9].

Similarly, numerous studies show that fruits and vegetables’ losses in ascorbic acid and B vitamins are lower during frozen storage [10,11]. A study comparing the loss of vitamin C in broccoli, carrots, legumes, peas, and spinach stored at refrigerator temperatures for 7 days and at deep freezing for 12 months showed that the losses were slighter in storage by freezing rather than by consuming the same fresh products [11]. The frozen products industry has advanced significantly, guaranteeing food safety and quality throughout the entire food chain [8]. Moreover, in competition with emerging technologies of minimum food processing, industrial freezing is the most suitable approach for keeping quality over lengthy storage periods [8]. According to Storey et al. [12], consumers of frozen fruit and vegetables consumed significantly more total fruit and vegetables than non-consumers of the same, enabling a higher intake of nutrients, such as dietary fiber, potassium, calcium, and vitamin D. Additionally, mean energy intake was considerably lower among children aged 1–18 years who ate frozen fruit and vegetables, whereas eating frozen fruit and vegetables did not affect adult energy intake. Salvadori and Mascheroni [13] showed positive commercial quality advantages of fast-freezing foods in energy consumption, throughput, and yield. However, no such actual commercial quality improvements were discovered in the research of Fernandez-Martin et al. [14].

Freezing is also as low cost (sometimes even lower) as other preservation methods [15]. However, as previously mentioned, frozen products are often seen as second-hand and not as high in quality as fresh produce, with low nutritional value, low healthiness, and less taste [16].

The current research aims to show Albanian and Kosovo consumers’ perception on FFs by focusing on three research questions (RQ). RQ1: What is consumer’s general perception in both countries regarding FF attributes, such as taste, nutritional value, and quality and food safety? RQ2: Does perception toward FF affect willingness to pay for fresh food? RQ3: Is there a demographic pattern of consumers preferring FF? The information provided will help the farming industry and policymaking follow the path to develop this sector further in both countries, and will lead in more sustainable agriculture by decreasing food loses and wastes.

The remainder of the paper is structured as follows. A literature review in the following section will present the main factors that affect consumer choices and behaviors while considering FF. In the third section, the methodology is presented. Discussion of results is presented in the fourth section. Conclusions and recommendations will end the paper.

## 2. Literature Review

Due to the changing consumer profile, the FF industry has changed significantly. The major trend in consumer behavior documented over the last half-century has been the increase in working women and the decline in family size. These two factors have resulted in a reduction in time spent preparing food. The entry of more women into the workforce also led to improved kitchen appliances and increased the variability of ready-to-eat or FFs available in the market [17]. Besides, the increased usage of microwave ovens affects food habits in general and the FF market, and allows rapid meal preparation and greater flexibility in meal preparation [3]. When analyzing consumer behavior, multiple-attribute frameworks are often dichotomized into intrinsic and extrinsic cues. This dichotomy of attributes is based on the approach of consumer economics presented in [18], in which products possess many characteristics [19] characterized by Darby and Karni [20] in their search for experience and extrinsic attributes. Intrinsic cues are integral and inseparable from the physical product, e.g., flavor, color, or texture for foods. In contrast, extrinsic cues are not physical components of the product, and changes have no material effects on the actual product, e.g., origin, price, packaging, health claims, brand name, etc. [21]. Based on this informational perspective, in analyzing FF, consumer perceptions studies are based on several attributes, such as health, food safety claims, quality, nutritional values, origin, taste, and trust in public agencies.

Consumers perceive ready-made FFs (RMFFs) as nourishing, healthier, and delicious, and are willing to pay the premium cost [22]. There are significant differences between countries, but, in general, consumers see health claims as useful; and they believe that the government should approve claims [23]. Consumers view food as healthier if it carries a health claim, and this “halo” effect may discourage them from seeking further nutrition information [23]. However, Ford et al. [24] show quite the contrary. Health claims do not influence the processing of nutrition information on a food label; health claims and nutrition information have independent effects on consumer beliefs. Lyly et al. [25] show that consumer willingness to pay for frozen soup is more affected by taste than the health claim. Another study on health-related claims shows that older individuals with limited income and ill health are the most affected by health claims on food [26]. Additionally, fresh fish was perceived as the healthiest fish product, followed by frozen, preserved, and ready-meal fish products [5]. According to Peavey et al. [4], it is essential to provide clear instructions for handling and to confirm that frozen fish is as nutritious as fresh fish.

Health claims seem to carry the message of increased healthiness for consumers. Still, they do not necessarily make the product more appealing [11]. However, including contextual influences and realistic conditions in assessing consumer understanding and use of health claims in purchase decisions is crucial because this claim can mitigate public agencies’ lack of trust in assuring food safety, especially in developing countries [12].

The prevalence rate of food-borne diseases is higher in low-income than in high-income countries. The use of unsafe water for cleaning and food processing, substandard food production processes and poor food handling, lack of adequate food storage facilities, and inadequate or poorly enforced food safety laws are some causes [27]. Studies show that freezing may be used in the future to reliably reduce populations’ food-borne pathogens, as well as to preserve food [28]. However, consumers lack information in this direction, and more education is needed. The quality of food products is usually associated with freshness, food safety, nutrition, and value [29]. Higher consumer perceptions of nutritional content, sensory appeal, and price lead to higher attitudes toward frozen meat and increased purchase intention [30].

Origin is also a widely considered attribute when dealing with FF and consumer behavior. Claret et al. [31] show that the country-of-origin information affected more choices when choosing fish than freshness/frozen information. A study conducted by Haas et al. [32] revealed that local origin, among expiry date, domestic, and brand reputation, is the most frequently used safety and quality cue for Kosovo and Albanian consumers. Other research has shown that the negative value of the country of origin could reduce the impact on the purchase behavior of imported FF [33]. The FF attribute competes with fresh food in another food context because fresh means local, and the latter translates into organic [16,34,35,36,37,38].

Lack of trust in food safety represents an alarming situation for the late-modern world because ensuring food safety is one of the primary tasks of human society [39]. Finstad [40] shows how the frozen fish industry created consumer trust despite initial skepticism. The authors list conventional ways of building trust, such as quality control systems, branding, marketing, and trust-producing technologies. The latter is a valuable lens for understanding how consumers become familiar with new foods. Other factors influencing consumers in developing countries to switch to alternative products such as FF include the loyalty to buy fresh food [16], the desire to maintain traditions [35,37], and the need to be in control and perceive higher informational value. A study undertaken in China indicates that the interplay of earlier factors motivates consumers to resist or embrace new consumption choices [41].

The Mediterranean diet pyramid based on Greek food patterns and Southern Italy in the early 1960s has been associated with good health [1]. Life expectancy was among the highest globally, and rates of coronary heart disease, certain cancers, and other diet-related chronic diseases were among the lowest compared to other European countries [1]. The Mediterranean diet is usually perceived as eating fresh food due partly to climate possibilities. A study with Turkish consumers shows higher preferences for fresh food, supported by the consumer claim that the climate characteristics allow the consumption of fresh food during all the seasons of the year, especially fruits and vegetables [42]. Another study shows that Portuguese consumers’ habits also prefer fresh products instead of frozen ones [43]. A cross-sectional study conducted with consumers in the Czech Republic, Germany, Greece, Italy, Portugal, Romania, Sweden, and the UK showed that fresh fish was perceived the healthiest fish product, followed by frozen, preserved, and ready-meal fish products. Perception scores were highly correlated with self-reported fish consumption in Mediterranean countries [2]. Climate factors are the factors that expose Mediterranean consumers to fresh food for the whole year. Food habits, while shaped by culture, are dynamic and susceptible to changes through a process of acculturation brought about by migration to a new country [3]. Albania and Kosovo are two countries that have experienced emigration throughout their existence. After the nineties, Albania was dominated by intensive migrant flows toward Greece, Italy, other EU countries, and the USA. However, there is a larger emigrant concentration in Italy and Greece for Albanians, while about 60% of Kosovo emigrants live in Germany and Switzerland [44]. Dietary changes and emigration are well-documented, showing that, upon immigrating to a new country, it is often difficult to maintain traditional eating habits. In a previous study on Quality of Frozen Food [3], the authors show that, when moving from the south of Europe toward the north, consumers tend to change their consumption habits toward a more westernised diet, favouring frozen food. However, the case of Albania is quite the contrary; the emigration toward Italy and Greece has reinforced the eating attitudes toward the Mediterranean diet. Following this assumption, and the Mediterranean climate that offers the possibility for fresh food, we can suggest that Albanian consumers will be more in favour of fresh food than those of Kosovo. Additionally, linked to the Mediterranean diet is the role of locality and origin in consumer preferences, as in other Mediterranean countries. Albanian consumers show a high inclination toward the product’s origin by considering the origin as a surrogate indicator for a safer food [4]. In contrast, consumers of Kosovo have been exposed more to a continental diet than to the Mediterranean one and they were always open to foreign markets, which is why the linkage with the origin is lower [32].

Consumer awareness and access to information and technological innovations for food freezing have changed consumers’ perceptions of these products. Influencing factors analysis on consumer behavior is necessary to inform the industry and policymakers to undertake targeted integrated communication to the consumer, by appealing to the pieces of information that are relevant for them, translating into a shorter decision and buying process.

The importance of this research lays in the following areas: (1) up to now, there is a lack of data covering consumer perceptions of frozen foods in Albania and Kosovo; (2) the importance of uncovering the main attributes that they appreciate most when buying frozen foods; (3) the importance of uncovering the factors that influence their decision-making process toward consumption of frozen foods; (4) determining how well informed they are about the nutritional values of frozen foods.

The information gained from this research may be useful for all the actors involved in frozen food production and marketing chain.

In the present study, attributes such as taste, nutritional value, health claim, and trust in storage, transportation, and origin are analyzed. The following section explains the roadmap followed to understand consumer perceptions of Albania and Kosovo.

## 3. Materials and Methods

### 3.1. Research Design

The data has been collected through consumers from Albania and Kosovo using a quantitative, structured survey and a previously tested questionnaire. A group of trained MA students conducted the interviews outdoors in marketplaces during March–June 2020. Surveys are one of the most popular approaches in social research and share three crucial characteristics: (1) wide and inclusive coverage; (2) capturing a specific point in time; (3) empirical research. The main aim of a survey is to provide information and bring things up to date; however, many researchers choose to present historical survey results to show how things used to be at particular point in the past [45]. Before starting the survey, two focus groups with experts in market research were undertaken in both countries. The questions directed to experts of both countries were the following: What are your general perceptions of FFs? Will you replace FFs with fresh food? No specific FFs were included in the analysis. This first step enabled us to identify the main attributes consumers’ value while considering FFs in their food choices. The main attributes considered by the consumers in both countries are grouped in the following categories: (1) quality and safety; (2) consumers information level; and (3) product origin. Identifying these attributes in the focus groups helped the development of the questionnaire used as an instrument to evaluate consumers’ attitudes toward FFs in a contextual, grounded way. The questionnaire is composed of three sections. In the first section, data on the demographics of the respondents, such as age, gender, level of education, income, civil status, and family size, were collected. The second section provides data on household food purchases and attitudes towards FFs. It consists of eight statements, and the measurement of attitudes toward FFs was assessed through the Likert scale from 1 = ‘strongly disagree’ to 5 = ‘strongly agree’.

### 3.2. Data

A total sample of 376 valid questionnaires was taken from this survey. The mode by which the questionnaire was administered can potentially influence consumer responses. In this study, we used the traditional ‘paper and pencil’ mode for data collection.

The face-to-face interview is considered to be one of the least burdensome methods, as the only requirement from the respondent is to speak the same language in which the questions have been asked, and own basic verbal and listening skills. No reading skills are required, in contrast to other modes of data collection that make greater auditory demands and may be burdensome to respondents. In addition, a survey with its face-to-face contact offers some immediate means of data validation in that the researcher can sense whether the respondent is providing false information [45,46]. A non-probability approach was used in respondents sampling, meaning that consumers in markets who buy FFs were selected from the population being studied. Researchers usually use three different approaches to the calculation of the sample size: statistical, pragmatic, and cumulative. In our study, we used the pragmatic approach, and this was partly because of the resources available and the difficulties in meeting respondents face-to-face due to the COVID-19 restrictions. The study targeted consumers of Prishtina and Tirana, which are the biggest cities in Kosovo and Albania in terms of population. These two cities represent the most attractive markets for food industry as the largest economic, administrative, educational, and cultural centers of these two countries are located in Prishtina and Tirana, including the concentration of purchasing power. Sociodemographic data of respondents are presented in Table 1 bellow.

The third section of the questionnaire consisted of the contingent valuation (CV) scenario. The CV method is based on a questionnaire that gathers information based on consumer declaration. It is considered a declarative method because consumer declarations are analyzed, rather than data from real situations [47]. Several researchers in the food sector have used this method to evaluate consumer preferences regarding the economic sacrifices they are willing to undertake [16,48,49]. The application of this method does not require the estimation of the tradeoffs between the attributes. In the present study, respondents were presented with five bids, and were asked the following: ‘assume that one kg of FFs product is priced at one Euro, how much more you would be willing to pay in change with fresh products?’ Interviewed consumers had an opportunity to choose one of the following alternatives: (a) 1 + 10% more = EUR 1.10, (b) 1 + 20% more = EUR 1.20, (c) 1 + 30% more = EUR 1.30, (d) 1 + 40% more = EUR 1.40, (e) 1 + 50% more = EUR 1.50, No payment WTP = EUR 0. The EUR 1 point represents an anchor price that helps the consumer’s decision process. A boxplot has been used to identify outliers or errors in each variable included in the analysis. Good hygiene of data was ensured prior to the data analysis. We used the t-test to identify statistical differences between two samples included in the study; r Spearman’s rho test and the bootstrapping method were performed to find out correlation between consumer perception on FFs and demographics. At the same time, the Mann–Whitney *U* test was calculated to test whether there is a difference between Kosovo and Albanian consumers regarding their WTP for FFs.

## 4. Results and Discussion

For assessing the perception of FF, the respondents were asked to specify their agreement using a 5-point Likert scale (1 = total disagreement with the statement; 5 = total agreement with the statement), with statements concerning the (1) quality and safety of FFs, (2) information on FFs, (3) impact of the origin of FFs on their perception of the food products. As shown in Table 2, consumers’ perception of FF regarding the first sub-session of statements on quality and safety of FFs, there is a statistically significant difference (*p* ≤ 0.001) between the two Balkan countries. In the case of Kosovo, all means are below the neutral (three), varying from 2.41 to 2.82, showing that consumers from Kosovo disagree with the statements (see Table 2). The results show that they have higher preference for FFs and consider them healthy with high nutritional values, good taste, and believe that they are properly stored with no quality and safety issues. Meanwhile, Albanian consumers show means of responses above the average of 3, varying from 3.30 to 3.94, showing statistically significant lower preferences toward FFs. Similar results are obtained with the third session of statements, showing a lower preference for imported FFs from Albanian than Kosovo consumers (*p*-value of ≤0.001). However, even local FF is not preferred due to the lack of trust in their safety. The majority of consumers (55%) consider quality to be one of the most important attributes when buying FFs.

Regarding the perceived quality and safety of FFs, the results show that Kosovo consumers display a higher means in all items under the construct ‘quality and safety of FFs’ than Albanian consumers (see Table 2). A similar tendency was observed for information and origin of FFs. Statistically, significant mean differences were observed except “I do not prefer frozen products for no particular reason”. The lack of information on FFs proves a need to provide and improve their information level regarding the FFs. According to Grunert et al. [50], distribution systems for food can change quickly in emergent markets, requiring new competencies for consumers. It has been observed that higher levels of competence lead to more information searches and better shopping outcomes for consumers [51].

The results in the case of Albania show that consumers consider FFs less healthy, nutritional, and tasty when compared with fresh foods. Other studies also show a common perception that food preservation or processing significantly reduces nutritional quality [34]. These perceptions continue to affect consumer decisions, especially in emerging economies. For emerging economies, especially in rural or semi-rural areas, the FFs industry did not develop significantly, and this might be related to the consumers’ perception and demand for FFs. In addition, consumers lack a trust in food safety institutions, which are responsible for ensuring adequate technology in the freezing process and storage [16]. Several studies have shown lower food safety and quality levels [32,52,53]. According to the study conducted by Hass et al. [32], expiry date, domestic and local origin, and brand reputation are the most frequently used safety and quality cues of consumers from Kosovo and Albania. Other international food standards, such as ISO or HACCP, are less frequently used as quality cues, showing a need to strengthen the institutional capacities related to food safety and quality following best practices from EU countries. The results show that Albanian consumers display a higher lack of trust in the food safety linked to FF. Similarly, in this category of food products, even buying local does not mitigate their fears. Other studies show Albanian origin in food products as a proxy of safer food than imported products [15].

The less positive perception towards FFs by Albanian consumers can be explained by their higher WTP (Table 3) for fresh foods when compared with Kosovo consumers. The Mann–Whitney *U* test proved a statistically significant difference between consumers of the two countries when questioning their WTP for fresh food as an alternative to FFs. Moreover, the results of positive perception towards FFs by Kosovo consumers can be supported with the higher frequency of stating “yes we do consume FFs” (χ^2^ = 6.20; df = 2; *p* ≤ 0.05) and their higher consumption frequency of FFs than Albanian consumers (Kosovo mean rank = 205.13; Albania mean rank 170.13; Mann–Whitney *U* test = 13984.5; *p* = 0. 001). According to the obtained results of this study, Kosovo consumers would be willing to consume even more if assurance for quality and safety of FFs by responsible authorities improves (χ^2^ = 23.85; df = 2; *p* ≤ 0.00).

In the case of Albanian consumers, as shown in Table 4. The age of respondents was proved to be positively and significantly correlated with the perception of quality and safety of FFs. This indicates that, with the increase in consumers’ age, positive perception towards FFs decreases. Considering that the average age in Albania is relatively low, we can assume that there is an opportunity to improve perception and attitudes towards FFs further if the institutions in charge assure quality and safety simultaneously. In the Kosovo case, age was not significantly correlated with any of the items grouped in the quality and safety constructs. Information for FFs was not shown to be significantly correlated with the age of consumers in Kosovo or Albania, whereas the origin of FFs was significantly correlated with the age of Albanian consumers, including Kosovo, for the item, “I do not buy local FFs because i do not trust their safety”, evidencing that safety is still an issue in both countries. 

The correlation analyses between consumers perception on frozen foods and their level of education, Table 5, shows that Education of Kosovo consumers was positively and significantly correlated with the perception that FFs are not healthy; Albanian consumers with a higher education considered fresh food tastier than FFs, and in both countries higher educated consumers perceive local FFs less trustful.

Kosovo consumers have higher preference in freezing food by themselves at home compared with consumers from Albania. Over 85% of consumers from Kosovo stated that they freeze food in their household, while for Albanian consumers this figure was around 32%. This difference was proven to be statistically significant (χ^2^ = 53.120, *p*-Value = 0.000).

## 5. Conclusions

The current research provides the following: (1) information on the perceptions of consumers from Albania and Kosovo on the quality and safety of frozen foods; (2) information on the level of knowledge on FFs; (3) information on the impact of origin on perceptions toward FFs and the correlation between (4) age and (5) education level in perceptions toward FFs.

Regarding the consumers’ perception of FFs on (1) the quality and safety of frozen foods, this study shows that there is a statistically significant difference (*p* ≤ 0.001) between the two Balkan countries. In the case of Kosovo, our results show that there is a higher preference for FFs, considering them to be healthy with high nutritional values and good taste, and consumers believe that they are properly stored and have no quality or safety issues. Albanian consumers show statistically significant lower preferences toward FFs. A statistically significant *p* value difference of 0.004 was obtained on their perception of their own knowledge level on FF’s, and Kosovo consumers consider themselves more informed about this category of products.

Similar statistical significance was obtained with the third session of statements, showing a lower preference for imported FFs among Albanian consumers compared with Kosovo consumers (*p*-value of ≤0.001). Furthermore, even local FF was not preferred by Albanian consumers due to the lack of trust in their safety, while there was a statistical difference of *p* value ≤0.001 on this issue among Kosovars.

The less positive perception towards FFs by Albanian consumers can be explained by their higher WTP for fresh foods when compared with Kosovo consumers (χ^2^ = 6.20; df = 2; *p* ≤ 0.05), and their higher consumption frequency of FFs compared with Albanian consumers (Kosovo mean rank = 205.13; Albania mean rank 170.13; Mann–Whitney *U* test = 13984.5; *p* = 0. 001). According to the obtained results of this study, Kosovo consumers would be willing to consume even more FFs if assurance for quality and safety from the responsible authorities improves (χ^2^ = 23.85; df = 2; *p* ≤ 0.00).

In the case of Albanian consumers, age was proved to be positively and significantly correlated with the perception of quality and safety of FFs, while for Kosovars, it was not found to have a significant correlation.

The Education level of consumers from both countries was negatively correlated with their perception on quality and safety of FFs.

A statistically important difference (χ^2^ = 53.120, *p*-Value = 0.000) was found in the question if the respondents freeze food products by themselves in their households with over 85% of Kosovars and 32% Albanians who responded positively to this question. 

The market of frozen foods in both countries is mainly dominated by imports. There is potential for the production of primary products and the development of freezing technologies in both countries that would have a direct impact on generating income for businesses involved in this chain, starting from local farmers.

Future studies on factors influencing consumer perceptions and consumer attitudes need to be conducted.

In both countries, responsible authorities should help the consumers to have a more profound knowledge concerning the safety and quality of FFs. Necessary measures will boost the consumers’ trust, increasing farmers’ incomes, and impacting sustainable agriculture, preventing food waste and losses. Applying a hypothetical scenario, such as the CV method, not directly to frozen food but to fresh food, is an essential methodological process, yielding valuable results that would have been difficult to collect with more complex methods, such as the choice experiment or experimental economics (auctions).

### Limitations

The study has some limitations with regard to the small sample size and, specifically, the very different distribution among the age groups. The small sample size had to be tailored to meet the constraints imposed by the COVID-19 situation. As the interviews took place in supermarkets, due to the higher risk for older consumers during the pandemic, younger consumers were more likely to go to the shops and were more present in the supermarkets. This could possibly be the main reason why the share of the group aged 18–34 is the highest one. In addition, both countries are characterized by a relatively young population. Another limitation of this study is the focus on frozen food consumption generally and not on a specific product. However, this was the first step in understanding the product categories that consumers prefer and the inclination toward frozen food. Future studies will be undertaken to understand other factors influencing both countries’ perceptions and behaviors toward frozen food. Assessment of the influence of acculturation on food habits due to emigration will be the next step in understanding whether emigration has affected food habits due to differences in the emigrant destinations of Albanians and Kosovars. Additionally, demographic changes such as changes in the gendered roles of household food preparation may have an effect. Additional studies are needed to explore whether the lack of trust in food safety institutions inhibits the successful development of FFs in Albania and Kosovo.

## Figures and Tables

**Table 1 foods-11-01247-t001:** Sociodemographic variables of the sample.

	Category	Kosovo % (*n* = 182)	Albania % (*n* = 198)
Age	18–24 years old	47.2	16.7
	25–34 years old	22.7	32.3
	35–44years old	23.3	18.7
	45–54 years old	5.7	20.2
	55–64 years old	1.1	9.1
	More than 65 years old	0.0	3.0
Gender	Male	10.2	32.3
	Female	89.8	67.7
Education	Basic–Middle School	1.1	1.0
	High School	27.3	17.7
	Higher Education, University	56.3	55.1
	Other type of education	15.3	26.2
Monthly income	Up to EUR 100	0.6	1.0
	EUR 101–200	2.8	6.6
	EUR 201–300	11.9	16.8
	EUR 301–400	22.6	25.4
	EUR 401–500	18.1	18.8
	EUR 501–600	17.5	31.5
	EUR 601–700	11.9	0.0
	EUR 700 or more	14.7	0.0

**Table 2 foods-11-01247-t002:** Comparison of consumers’ perception on FFs between Kosovo and Albania.

Quality and Safety of FFs	M Kosovo	M Albania	MD	t	*p*	MD 95% Confidence Interval	
I do not prefer FFs as they are not healthy	2.41 (1.41)	3.49 (1.42)	−1.080	−7.317	≤0.001	−1.370	−0.789
FFs contain less nutritional value than fresh produce	2.72 (1.56)	3.94 (1.34)	−1.223	−8.099	≤0.001	−1.520	−0.926
FFs are not as tasty as fresh products	2.80 (1.47)	3.77 (1.39)	−0.969	−6.518	≤0.001	−1.261	−0.676
I do not prefer FFs because they are not stored in the right conditions, long time in the refrigerator, freezing/thawing possible during the distribution chain	2.82 (1.27)	3.30 (1.47)	−0.481	−3.367	≤0.001	−0.763	−0.200
Information for FFs							
I do not prefer FFs as I do not have much information about them	2.39 (1.16)	2.77 (1.36)	−0.381	−2.894	0.004	−0.640	−0.122
I do not prefer FFs for no particular reason	2.59 (1.29)	2.54 (1.40)	0.050	0.361	0.718	−0.225	−0.326
Origin of FFs							
I do not buy FFs imported products because I am not sure about their quality	2.70 (1.45)	3.24 (1.60)	−0.540	−3.392	≤0.001	−0.853	−0.227
I do not buy FFs imported products because they are not healthy	2.51 (1.46)	3.25 (1.45)	−0.740	−4.886	≤0.001	−1.039	−0.442
I do not buy local FFs because I do not trust their safety	2.41 (1.40)	3.32 (1.55)	−0.908	−5.900	≤0.001	−1.211	−0.605

**Table 3 foods-11-01247-t003:** Respondents WTP for fresh foods instead of FFs.

	Kosovo WTP (%)	Albania WTP (%)
1 + 10% more = EUR 1.10	21.5	7.6
1 + 20% more = EUR 1.20	15.2	13.2
1 + 30% more = EUR 1.30	22.2	19.2
1 + 40% more = EUR 1.40	11.4	12.6
1 + 50% more = EUR 1.50	29.7	21.2
100+ %	0.0	26.3

Mann–Whitney *U* test = 10549.000; *p* ≤ 0.000; mean rank for Kosovo = 146.27; mean rank for Albania 204.22.

**Table 4 foods-11-01247-t004:** Correlation coefficients between age and consumers’ perception for frozen foods in Kosovo and Albania.

Statements	Location	*r* Sperman-Rho	SE	*p*	Bootstrap r 95% Confidence Interval	
Quality and safety of FFs						
I do not prefer FFs as they are not healthy	KS	0.044	0.079	0.562	−0.110	0.198
	AL	0.229 **	0.051	0.000	0.128	0.326
FFs contain less nutritional value than fresh produce	KS AL	0.112 0.173 **	0.077 0.051	0.141 0.001	−0.042 0.072	0.261 0.271
FFs are not as tasty as fresh products	KS AL	−0.033 0.177 **	0.078 0.050	0.662 0.001	−0.186 0.078	0.116 0.274
I do not prefer FFs because they are not stored in the right conditions, long time in the refrigerator, freezing/thawing possible during the distribution chain	KS AL	−0.141 0.104 *	0.077 0.051	0.062 0.044	−0.294 0.004	0.008 0.206
Information for FFs						
I do not prefer FFs as I do not have much information about them	KS AL	−0.045 0.020	0.082 0.054	0.552 0.708	−0.204 −0.089	0.118 0.123
I do not prefer FFs for no particular reason	KS AL	0.068 0.004	0.079 0.052	0.372 0.938	−0.079 −0.099	0.225 0.106
Origin of FFs						
I do not buy FFs imported products because I am not sure about their quality	KS AL	0.013 0.172 **	0.077 0.052	0.861 0.001	−0.136 0.071	0.166 0.274
I do not buy FFs imported products because they are not healthy	KS AL	0.071 0.163 **	0.078 0.051	0.349 0.002	−0.083 0.059	0.224 0.261
I do not buy local FFs because I do not trust their safety	KS AL	0.185 * 0.227 **	0.076 0.051	0.014 0.000	0.031 0.123	0.332 0.324

Note: * *p* < 0.05; ** *p* < 0.001; KS—Kosovo; AL—Albania.

**Table 5 foods-11-01247-t005:** Correlation coefficients between education and consumers’ perception for frozen foods in Kosovo and Albania.

Statements	Location	r Sperman-Rho	SE	*p*	Bootstrap r 95% Confidence Interval	
Quality and safety of FFs						
I do not prefer FFs as they are not healthy	KS AL	0.191 * 0.073	0.075 0.055	0.011 0.161	0.041 −0.034	0.335 0.182
FFs contain less nutritional value than fresh produce	KS AL	0.101 0.060	0.075 0.054	0.185 0.247	−0.046 −0.046	0.248 0.163
FFs contain less nutritional value than fresh produce	KS AL	0.044 0.109 *	0.078 0.053	0.562 0.036	−0.110 0.006	0.195 0.211
FFs are not as tasty as fresh products	KS AL	0.142 0.076	0.081 0.055	0.059 0.142	−0.021 −0.031	0.298 0.184
Information for FFs						
I do not prefer FFs as I do not have much information about them	KS AL	0.122 0.046	0.076 0.053	0.107 0.380	−0.028 −0.056	0.270 0.149
I do not prefer FFs for no particular reason	KS AL	0.087 0.020	0.078 0.053	0.253 0.708	−0.064 −0.083	0.241 0.126
Origin of FFs						
I do not buy FFs imported products because I am not sure about their quality	KS AL	0.009 0.033	0.075 0.052	0.907 0.521	−0.140 −0.069	0.156 0.135
I do not buy FFs imported products because they are not healthy	KS AL	0.061 0.066	0.078 0.054	0.419 0.206	−0.091 −0.039	0.215 0.172
I do not buy local FFs because I do not trust their safety	KS AL	0.236 ** 0.220 **	0.072 0.051	0.002 0.000	0.094 0.124	0.374 0.319

Note: * *p* < 0.05; ** *p* < 0.001; KS—Kosovo; AL—Albania.

## Data Availability

The data presented in this study are available on request from the corresponding author. The data are not publicly available due to privacy.

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
