# Peer review of "Analyzing Consumer Perception on Quality and Safety of Frozen Foods in Emerging Economies: Evidence from Albania and Kosovo"

_foods, 2022, doi:10.3390/foods11091247_

Round 1
Reviewer 1 Report
The research was surveyed research and was done in 2020.
Is it still relating to now consumer perception?
The similarity index was good enough (checked by Turnitin).
But the scientific soundness "is not enough based on my point of view" because it only informs the readers about the consumer's perceptions and attitudes.
I hope the authors could add more scientific analysis about quality and safety.
Author Response
Survey is one of the most popular approaches in social research and share three crucial characteristics: 1) wide and inclusive coverage; 2) at a specific point in time; 3) empirical research. The main aim of a survey is to provide information and bring things up to date, however many researchers choose to present historical survey results to show how things used to be at particular point in the past.
We have been analyzing consumers’ perception on quality and safety of frozen foods. The items included in the construct quality and safety of FFs (I do not prefer FFs as they are not healthy; FFs contain less nutritional value than fresh produce; FFs are not as tasty as fresh products; I do not prefer FFs because they are not stored in the right conditions, long time in the refrigerator, freezing/thawing possible during the distribution chain; Table 2 in the manuscript) present scientific analysis about the quality and safety. In addition we added results from the question of how do consumers evaluate quality attributes when buying FFs.
Reviewer 2 Report
The aim of the paper is to compare Albanian and Kosovo consumers’ perception on frozen food quality and related attributes such as taste, nutritional value, food safety. The topic can be interested but there are several critical points that need to be fixed:
The connection between the literature review and the hypotheses developed is not clear. All the hypotheses should be more clearly related to a corresponding literature, highlighting on which area the current knowledge lacks.
The methodology description should be more in-depth presented (questionnaire administration? sample selection? data treatment? etc.).
The differences between the two samples should be discussed. Specifically, the very different distribution of age groups is a big limitation of the study which should be explained as regards the criteria for sampling selection adopted, and it should be underlined in the limitations of the study.
Conclusions are very generic and they don’t make clear how the study contributes on knowledge in the field analyzed. Similarly, the practical implications are generic and should be better identified. If these two issues are not overcome the contribution of the paper cannot be identified.
Author Response
The Hypothesis are revised and presented in Introduction part as suggested by the guidance for authors https://www.mdpi.com/journal/foods/instructions .
The mode by which the questionnaire was administered can potentially influence consumer responses. In this study we used traditional mode ‘paper and pencil’ for data collection.
The face-to-face interview is considered to be one of the least burdensome methods as the only requirement from the respondent is to speak the same language in which the questions have been asked, and own basic verbal and listening skills. No reading skills are required, in contrast to other modes of data collection that make greater auditory demands and may be burdensome to respondents. In addition survey with its face-to-face contact offers some immediate means of data validation in a way that researcher can sense if the respondent is providing false information [45], [46].
Box-plot has been used to identify outliers or errors in each variable included in the analysis. Good hygiene of data was ensured prior to the data analysis.
The study has some limitations with regard to the small sample size and specifically, the very different distribution among the age groups. The small sample size had to be tailored to meet the constraints imposed by the Covid19 situation. As the interviews were place in supermarkets, due to the higher risk for older consumers in pandemic time, younger consumers were prone to do shopping and be more present in the supermarkets. This could possibly be the main reason why the share of group age 18-34 is the highest one. In addition, both countries are characterized with relatively young population.
Conclusions revised accordingly.
Reviewer 3 Report
Comments to the Editor and Authors
The manuscript Analyzing consumer perception on quality and safety of frozen foods in emerging economies: evidence from Albania and Kosovo seems to be interesting. However, I have a few comments:
i) Comparison of consumer’s perception across the two countries are not reliable with these small sample sizes. On the other hand, it is important to explain how these samples were obtaining and how they can be representive of the population. If the authors' option was a non probability sampling they should explain how and why they use these for the research. I think that these data could be of interest if presented in a different way but the manuscript needs to be widely revised/rewritten, explaining how the sample size is appropriate for the analysis. Also the limitations of using a non probability sampling should mentionned in the limitations section (6.2).
ii) The authors don't analyze the consumer’s attitudes of the two Balkan countries towards FFs but only their perceptions. Thus, Isuggest revising the aim by limiting it at “to show consumers’ perception”, considering the socio-demographics, (or other variables measured) associated with these perceptions. It means review the three research questions, aligning it with the research results.
iii) The conclusion must be improved with the clearly inclusion of the way in which the results obtained answer the research questions and how show that FF represent an alternative to small producers in developing countries such as Kosovo and Albania to add value to their products in a competitive market.
iv)A section or paragraph “Directions for future research” should be also added before the conclusion.
On the basis of detailed comments sent to the authors, I propose a major revision of the article prior.
Author Response
Comment i) relates to the comments from previous reviewer and it has been addressed in the methodology section.
ii) Addresses: it was deleted the “attitudes” part.
iii) Revised accordingly.
iv) It is also not part of the Research Manuscript Section according to the MDPI FOODS instruction for authors: https://www.mdpi.com/journal/foods/instructions. Therefore, it is added in conclusion part.
Round 2
Reviewer 1 Report
The authors have answered and improved the significance of content and scientific soundness.
Reviewer 2 Report
The authors improved the manuscript according to the reviewers' comments